# The Potential of Nail Mini-Organ Stem Cells in Skin, Nail and Digit Tips Regeneration

**DOI:** 10.3390/ijms22062864

**Published:** 2021-03-11

**Authors:** Anna Pulawska-Czub, Tomasz D. Pieczonka, Paula Mazurek, Krzysztof Kobielak

**Affiliations:** Laboratory of Stem Cells, Development and Tissue Regeneration, Centre of New Technologies (CeNT), University of Warsaw (UW), 02-097 Warsaw, Poland; a.pulawska-czub@cent.uw.edu.pl (A.P.-C.); t.pieczonka@cent.uw.edu.pl (T.D.P.); p.mazurek@cent.uw.edu.pl (P.M.)

**Keywords:** nail organ, nail stem cells (NSCs), nail proximal fold stem cells (NPFSCs), nail matrix, nail regeneration, skin regeneration, digit tip regeneration, BMP signaling, WNT signaling

## Abstract

Nails are highly keratinized skin appendages that exhibit continuous growth under physiological conditions and full regeneration upon removal. These mini-organs are maintained by two autonomous populations of skin stem cells. The fast-cycling, highly proliferative stem cells of the nail matrix (nail stem cells (NSCs)) predominantly replenish the nail plate. Furthermore, the slow-cycling population of the nail proximal fold (nail proximal fold stem cells (NPFSCs)) displays bifunctional properties by contributing to the peri-nail epidermis under the normal homeostasis and the nail structure upon injury. Here, we discuss nail mini-organ stem cells’ location and their role in skin and nail homeostasis and regeneration, emphasizing their importance to orchestrate the whole digit tip regeneration. Such endogenous regeneration capabilities are observed in rodents and primates. However, they are limited to the region adjacent to the nail’s proximal area, indicating the crucial role of nail mini-organ stem cells in digit restoration. Further, we explore the molecular characteristics of nail mini-organ stem cells and the critical role of the bone morphogenetic protein (BMP) and Wnt signaling pathways in homeostatic nail growth and digit restoration. Finally, we investigate the latest accomplishments in stimulating regenerative responses in regeneration-incompetent injuries. These pioneer results might open up new opportunities to overcome amputated mammalian digits and limbs’ regenerative failures in the future.

## 1. Dissection of Nail Mini-Organ Anatomy and Function

Nails are the skin appendages located at the distal phalanx of each finger and toe of the human body, providing physical protection of the dorsal tips and assistance in fine manipulations and subtle finger functions, including scratching. The structure begins at the eponychium, separating a thickened layer of skin epidermis from the nail organ, which firmly adheres to the nail plate and protects the area between the nail and epidermis from exposure to germs and contamination. At this point, the skin epidermis folds inwards ventrally, forming the nail proximal fold (NPF), which is histologically different from normal skin epidermis due to a ceasing epidermal differentiation, manifested by a loss of the granular layer with loricrin expression (Figure 1) [1]. As the wedge-shaped NPF bends once more, it attaches to the root nail plate’s dorsal surface forming the nail matrix towards the digit’s tip. The thick proliferating nail epithelium zones are profusely connected with nerves and are supplied with lymph and blood vessels. These actively proliferating nail keratinocytes are called the onychocytes, which establish the eosinophilic keratogenous zone (KZ). Next, the KZ differentiates into corneocytes, forming a protective, strong nail plate (NP) (Figure 1). During the maturation process, onychocytes broaden and flatten out, depositing to the NP. Once their nuclei disintegrate, onychocytes become corneocytes with AE13-positive hard keratins’ expression. The loss of nuclei contributes to the nail plate’s translucency, allowing the nail matrix at the NP’s base to be perceived in the form of a whitish crescent-shaped area called the lunula (Figure 1). The NP’s edges are surrounded by nail grooves and lateral nail folds. The hyponychium seals the plate to the finger at the distal end border and the beginning of the volar epidermis, preventing pathogens and pollution from entering underneath the plate (Figure 1). The folds under the surface of the NP, spanning between the end of the distal nail matrix (edge of the lunula) to the hyponychium, firmly attach to the complementary longitudinal epidermal ridges of the underlying nail bed (NB) and fasten the nail plate to the surface (Figure 1) [2]. The thin epithelium of NB consists of one or two layers of suprabasal postmitotic keratinocytes.

The visual distinctions between the digits of different mammals are apparent, as their function and evolution dictate the shapes of the nails and claws. However, recent studies have demonstrated that these units’ inner structures share major characteristics, especially human and mouse nail organs, with the latter already successfully employed as a model for the homologous human appendage [3]. Although the longitudinally curved murine NP, terminated with a sharp tip, is visibly distinguishable from a relatively flat and round-ended human nail, both organs consist of similar structures, including the NPF with eponychium, nail matrix along with KZ, NB, NP and hyponychium [4]. The base of the ventral surface of the NPF gives rise to the thick nail matrix (Figure 2), dorsally coated with KZ (Figure 1). Both layers suddenly become thin as they merge with the NB. A thin layer of loose connective tissue, rich in fibroblasts, vasculature and nerves, separates the nail epidermis from the triangular-shaped bone. Mouse NP encircles the distal phalanx from dorsal and lateral sides while leaving the exceptionally extended hyponychium uncovered (Figure 1).

Furthermore, to both structures sharing major characteristics, most keratins’ expression patterns are similar in both human and mouse digits [3]. These resemblances make the mouse nail mini-organ a perfect model to study nail biology and investigate digit regeneration mechanisms. Moreover, a wide variety of genetically modified mice enables researchers to analyze the genetic-based nail disorders and discover ways to overcome amputated mammalian digits and limbs’ regenerative failures. Ultimately, the results obtained from studies performed on mouse models might soon translate into a potential clinical setting in humans.

## 2. Searching for the Nail Mini-Organ Stem Cells

For their self-renewal and differentiation ability into the various specialized cell types, the adult stem cells (SCs) are responsible for upholding the homeostasis of tissues and organs by participating in their growth, renewal and regeneration. Each organ holds a specialized microenvironment, known as the niche, which helps to maintain the quiescence and regulate the proliferation and differentiation of the inherent stem cells [5,6]. The discovery of slow-cycling or quiescent label-retaining cells (LRC) allowed for identifying several skin stem cells across the epithelium in hair follicle bulge, cornea limbus, or sweat glands [7,8,9,10,11,12,13]. The BrdU pulse-chase procedure has been used in mice to identify the LRCs existence in their nail organ. However, that initial discovery has incorrectly localized those slow-cycling LRCs to the nail matrix’s basal layer adjacent to the nail bed [14]. This observation stood in contradiction to discoveries based on the lineage tracing of keratin 14 positive (K14^+^) basal epidermal cells, labeled with LacZ, which defined the proximal nail matrix as a collection of highly proliferative keratinocytes without LRCs [15]. It was also consistent with previous studies, which used radioactive tritiated glycine to predominantly label nail matrix [16,17]. Moreover, Sellheyer and Nelson’s complementary studies [18] focused on characterizing a slowly proliferating nail cells population within the NPF. During the embryogenesis, these cells expressed the same markers as the well-known hair follicle stem cells: keratin 15 (K15), keratin 19 (K19) and pleckstrin homology like domain, family A, member 1 protein (PHLDA1) [19] (Figure 3). Thus, in the beginning, the similarity between both cell types suggested further the existence of LRCs in the nail organ, however, localized rather in the ventral proximal fold instead of the previously proposed nail matrix. Finally, the recent studies, which used the doxycycline-regulated H2B-GFP LRC system in the *K5^TetOff^TreH2BGFP* transgenic mouse, clarified these discrepancies further and managed to address the previous inconsistencies. The LRCs were confirmed to be organized in a ring-like formation around the nail root in the nail proximal fold’s basal layer [20].

A few years later, Lehoczky and Tabin [21] identified the Lgr6 receptor (leucine-rich repeat-containing G protein-coupled receptor 6), a recognized marker of several epithelial SCs populations, in the murine nail stem cells (NSCs) of the nail matrix (Figure 3) and in a small cell subset across the distal digit bone and the eccrine sweat glands within the toe pad. In contrast, the presence of Lgr5 receptors was determined only in a unique dermal population of cells adjacent to the slow-cycling LRCs in the NPF and at the distal groove of the digit (Figure 3, marked by light blue) [20,21]. However, the role of these Lgr5^+^ cells in nail growth and homeostasis has yet to be defined.

## 3. Two Pools of Nail Mini-Organ Stem Cells

Hence, two different pools of stem cell population were identified within the nail mini-organ: the highly proliferative Ki67^+^ cells at the proximal matrix region and the slow-cycling LRCs (H2BGFP^+^) in the nail proximal fold (Figure 3) [20,22]. Between both factions, a gradient of less proliferative cells, labeled by both the Ki67^+^ and weak H2BGFP^+^ expression, marked the intermediate zone (IZ). The in vivo lineage tracing experiments in transgenic *K15^CrePR^Rosa26^LacZ^* mouse has shown that the slow-cycling K15^+^ nail proximal fold Stem Cells (NPFSCs) contribute to both the peri-nail epidermis and the nail plate structure, thus possessing bifunctional stem cells characteristics. Under physiological conditions, these cells are more involved in supporting the peri-nail epidermis tissue than the NP. However, following the injury, the homeostatic balance tilts toward the nail regeneration, and the NPFSCs adapt to the new circumstances by delivering the progeny to the nail matrix and differentiating into AE13-positive hard keratins of the nail plate [20].

On the other hand, the pool of NSCs discovered in the nail matrix, characterized as highly proliferative Ki67^+^ progenitor cells, was the main contributor to the external NP (Figure 2). These cells were located and described by Takeo et al. [15], who used a lineage tracing system in the *K14^CreER^R26R^LSL-LacZ^* transgenic mouse in order to mark a small subset of keratinocytes in the basal layer of skin and nail epidermis. Through a controlled expression of LacZ, the K14^+^ cells that took part in the nail growth were marked in the nail matrix and the nail bed. During 5 months of the experiment, the LacZ^+^ progenies were perceived as streaks in the NP, spreading linearly and distally from the proximal matrix. Apart from the expression of highly proliferative Ki67 marker and keratin 14 (K14), further analyses determined that proximal matrix cells expressed keratin 17 (K17) and possessed a high colony-forming ability observed in vitro. This feature confirmed that the proximal matrix indeed contained self-renewing NSCs that sustain nail growth. In comparison, no LacZ^+^ labeled cells were observed to emerge from the distal matrix.

The gene expression evaluation, performed between the distal and proximal matrix, characterized a major of the proximal NSCs with a downregulated Wnt signaling pathway, suggesting a direct involvement of the Wnt signaling in the nail differentiation [15]. The subsequently identified presence of the mediator of the Wnt signaling, the Lgr6 receptor, strongly supported this theory. Indeed, the genetic lineage-tracing experiments determined that the Lgr6^+^ cells contribute to the nail homeostatic growth, indicating Lgr6-marked cells as NSCs [21].

In summary, the nail mini-organ holds two distinct populations of stem cells, located in the nail matrix and the nail proximal fold. During the physiological nail growth, the matrix pool continuously delivers cells to the nail plate, while cells localized in the NPF predominantly support the peri-nail epidermis. The bifunctional characteristic of the NPFSCs becomes apparent upon a nail injury when the progenies from the NPF hasten to enrich the nail matrix and contribute to the regenerating nail plate.

## 4. Nail Growth, Differentiation and Homeostasis

The development of the nail organ in the human embryo begins around the 10th week of gestation [1]. At the 14th week, NP emerges from NPF and by the 17th week, the whole NB’s surface is covered by the NP. The fingernail of a healthy adult takes around 6 months to grow entirely, while this time is doubled for toenails [23]. It is generally acknowledged, that the nail matrix is predominantly accountable for the replenishing of the NP, while the bifunctional NPF cells can also contribute to the process; however, there is still an ongoing discussion whether other elements of the nail organ take part in the NP growth, especially the nail bed. A study performed by Zaias and Alvarez [16] confirmed that the nail matrix cells move superficially into the nail plate and distally into NB. The continuous replacement of the nail matrix is ensured by the cells’ divisions at its basal layer, which move the differentiating cells up, to be deposed through a KZ to the nail plate. At the same time, the pressure of accumulated cells moves the NP dorsally. However, based on the thickness and mass of the nail plate, the relatively inactive nail bed cells were also proposed to contribute a few horn cells to the distal nail plate’s ventral surface, possibly facilitating the distal movement as it grows [4,24]. Indeed, NB was observed to comprise cells expressing Msx1, a marker associated with multipotent and relatively undifferentiated progenitor cells, active within the developing and regenerating environments [25,26]. The minor input of the NPFSCs to the NP during the normal nail homeostasis was described by Leung et al. [20]. Further, the authors investigated these slow-cycling cells’ responsiveness upon a mechanical NP plucking on the *K15^CrePR^Rosa26^Tom^* mouse at the postnatal day 7 (P7). The keratinized structure entirely grew back within 2 weeks. The linear streaks of Tom^+^ cells (marked by Tomato) confirmed that the K15^+^ NPFSCs actively delivered progenies to the nail matrix and differentiated into the NP. NPFSCs also participated in the elongation of the NPF, providing extended protection of the exposed tissue. The following engraftment procedure of the H2BGFP^+^ nail LRC strip directly underneath the NPF of the immunocompromised NOD SCID donors confirmed the regenerative ability of the NPF LRCs [20]. Within several days, the engrafted cells were observed to successfully integrate with NPFSCs and contribute to the growing nail while retaining their slow-cycling physiology after an additional chase period. These findings established that normally quiescent NPFSCs are bifunctional and are able to adapt to the changing environmental circumstances by contributing to the nail restoration upon injury.

### 4.1. BMP Signaling Instruct Nail Mini-Organ Stem Cells to Differentiate into Nail Plate

The gene expression profiling of the nail slow-cycling LRCs, isolated from the NPF, revealed two inhibitors of the bone morphogenetic protein (BMP) signaling pathway, Decorin and Bambi, being down-regulated, thus signifying the role of the BMP pathway in nail homeostasis [20]. The canonical BMP pathway activity was verified through immunostaining against phosphorylated transcription factors of Smad1/5/8 (pSmads), revealing their expression in the NPFSCs and the nail matrix cells. Indeed, previous studies confirmed that Msx2 and Foxn1, the BMP downstream transcription factors, were required for the proper onychocyte terminal differentiation and the NB and KZ organization [27]. The *Msx2* mutants were characterized by poorly differentiated cells in the KZ and decreased production of the hard keratins. Even lower expression of hard keratins, manifested by broken nails, was observed for the *Foxn1* mutants. Additionally, double mutants developed hyperplasia of the nail bed [27]. In comparison, the deletion of the BMP receptor A1 (*Bmpr1a*) in the developing skin of the *K14^Cre^* transgenic mouse was correspondingly associated with irregular and thinner NP structure, hyperplastic NB, loss of the KZ and a reduction of the nail matrix cells’ proliferating abilities [20]. Moreover, the *Bmpr1a*-deficient nails were observed to adopt an epidermal fate, manifested with the extension of the skin epidermis granular layer throughout the nail structure. A similar observation was made for Hoxc13 and Foxn1 single mutants [28,29]. Additionally, the expression of keratin 1 (K1) and loricrin epidermal markers in the NP were identified instead of the hard keratin marker AE13 (Figure 3) [20].

### 4.2. WNT Signaling Activation in Distal Nail Matrix Essential for Nail Plate Formation

The Wnt signaling pathway is recognized to regulate the development of limbs and nail organs in embryos [30] and is required to restore lost limbs of vertebrates, which are known to retain significant regenerative abilities throughout adulthood [31]. The direct involvement of Wnt signaling in nail differentiation was thoroughly analyzed by Takeo et al. [15], who studied the population of highly proliferative NSCs in the nail matrix. A subset of cells located in the proximal region lacked the Wnt signaling mediator Tcf1 and the Wntless (Wls) protein. In contrast, in the distal part of the matrix, several keratins that contained TCF1 and LEF1 consensus binding sites were observed to be upregulated [15]. The evidence of canonical Wnt signaling in the nail epithelium was subsequently confirmed by Lehoczky and Tabin [21], who located markers of the Wnt pathway, TCF/Lef and Axin in the nail matrix of *TCF/Lef^H2B-GFP^* and *Axin2^LacZ^* transgenic mice. Moreover, they also identified the expression of Wnt pathway mediators: Lgr5 and Lgr6. Lgr5 was found in the mesenchymal cells adjacent to the nail proximal fold epidermis and across the distal groove, whereas Lgr6 expression was observed throughout the nail matrix, in a distal digit bone and in a toe pad (Figure 3). The Lgr receptor, in the presence of R-spondin (RSPO), forms an Lgr-R-spondin complex, which prevents the constitutive degradation of the Wnt receptors, thus making the Lgr-expressing cells ultimately more responsive to the Wnt signaling. During the normal digit and nail growth, marked cells of the *Lgr6^EGFP-ires-creERT2^*; *R26R^CAG-LSL-tdTomato^* heterozygous mice were observed to contribute either to the nail plate, digit bone, toe pad epidermis, eccrine sweat glands and ducts [21]. Nail formation in the conditional knockout mice models *K14^CreER^Wntless^fl/fl^* and *K14^CreER^β-catenin^fl/fl^*, which were deprived of either Wls or an essential mediator of the Wnt signaling, the β-catenin, was visibly abrogated [15]. The entire nail epithelium lacked the AE13 marker of keratinized nail cells and showed expansion of the NSCs along with the proximal matrix region, expressing a highly proliferative Ki67 marker and K17 (Figure 3). Moreover, the epithelial-specific β-catenin-deficient mice were also characterized with digit bone defects, as all the analyzed terminal phalanxes of the *K14^CreER^β-catenin^fl/fl^* mice were shorter and terminated with a pitted, uneven surface instead of a sharp-pointed tip [32]. As the cells in the osteolineage do not express the K14, this study confirmed the crucial role of the nail epithelium cells in maintaining the homeostasis of the underlying digit bone, modulated by the Wnt/β-catenin signaling components. Additionally, several studies have determined the essential role of the R-spondin family member protein, implicated in the Wnt pathway signaling in human nail development [33,34,35,36]. The mutation in the RSPO4 gene, encoding R-spondin 4, was associated with rare conditions: anonychia and hyponychia, manifested with absence or severe hypoplasia of fingernails and toenails. Affected individuals were not characterized either with deformations of the distal phalanx or defects in NB; therefore, the authors assumed that presence of R-spondin 4 is vital in rather advanced phases of embryonic nail development and/or later–in natural homeostasis of the organ.

In summary, the nail mini-organ development and homeostatic growth are precisely controlled by the BMP and Wnt signaling pathways. Genetically engineered mice models, deprived of active signals or their essential mediators, were characterized with visibly abrogated nails. The deformation was characterized by a lack of hard nail plate and loss of keratinization marker AE13. Instead, without BMP signaling, hard nail keratins were replaced by extended epidermal keratins, covering the entire nail organ. Moreover, the visibly reduced layer of the nail matrix no longer differentiates into the KZ. In contrast, the Wnt pathway’s deactivation manifested in an expansion of the proximal matrix and replacement of hard nail structure with sections of highly proliferative nail epithelium.

## 5. Digit Regeneration

Tissue regeneration is a complex process that aims to re-establish the polarity, structure and functionality of the disturbed or damaged fragment of the organ [37]. The digit tips of rodents and primates are recognized for their endogenous regeneration capability, including the digit’s nail plate, epidermis, nerves and bone. Nevertheless, this capacity was observed to be limited to the region adjacent to the nail organ area [38,39]. The distal amputation, which ensures consequent complete regeneration, removes around 23% of the distal phalanx length and 15% of the bone volume but leaves the bone marrow (BM), proximal nail matrix and footpad intact [40]. The extended amputation executed past this proximal boundary does not trigger an analogous regenerative response (Figure 4). This observation indicates that the nail epithelium and particularly the NSCs of the proximal nail matrix have a crucial role in orchestrating the complete digit tip regeneration.

The regeneration of an amputated digit tip is a multistep process, which commences with an inflammation of the damaged tissue as it becomes infiltrated by macrophages and neutrophils [41]. At this point, as the blood vessels are opened, the soft tissue of the injured site swells and a fibrin clot forms over the cut area [25]. During the following phase, called histolysis, the extracellular matrix, including the bone’s surface, is enzymatically degraded and erodes into two segments, ultimately exposing the bone marrow [39]. Accountable for this erosion response are the osteoclasts, which were observed to increase their volume directly after the amputation and during the histolysis process. The timing of the bone degradation in mice, studied by Fernando et al. [42], varied between 9–12 days following the injury [43] and strongly related to the completion of the wound closure, which was visibly inhibited by the remaining stump bone [44]. At the beginning of the histolysis process, the digit’s upper epidermis attaches to the lateral rim of the bone at the amputation level. Simultaneously, the ventral site’s tissue retracts proximally to connect to the external bone surface at the border of the emerging bone marrow cavity. The progressive degradation of roughly 50% of the distal phalanx bone volume provides a migration route for the epidermal cells. Newly formed transient wound epidermis (WE) separates the eroded distal bone stump, protruded into the clot (Figure 4). The WE is required for the digit blastema to establish and develop. For the secreted chemoattractant, stromal cell-derived factor 1 (SDF-1), WE acts as a signaling center for blastema cells. These cells are recognized to predominantly express the chemokine receptors type 4 and 7 (CXCR4/7) and pigment-epithelium derived factor (PEDF) receptors for the SDF-1 (Figure 4) [40,45]. Additionally, apart from facilitating cell migration, SDF-1 also mediates cell adhesion and survival [46].

The wound closure, which is now proximal to the original level of the amputation, outlines the blastema—an avascular [47], but innervated structure, which expanses from the exposed bone marrow cavity; the source of BM-derived stem cells and bone progenitor cells [40,42]. The origin of many blastema mesenchymal cells is still in debate, but their sources are being gradually uncovered [48]. The latest lineage tracing analysis determined that 26% of blastema cells originate from bone-associated cells, including the periosteum [40,49], as they express the Dmp1 (Dentin matrix acidic phosphoprotein1) marker for osteoblasts and osteocytes [50]. The connective tissue (CT) unquestionably plays a critical role in appendage regeneration by providing many cells to the blastema and by concealing the positional identity of the limbs [51]. The synthesis of the extracellular matrix (ECM) of the CTs is supplied by stromal mesenchymal progenitors (MPs), essential in modeling and/or remodeling the tissues’ structural integrity. Although the MPs display functional specialization, based on their tissue of origin and localization, it was long suspected that different types of MPs are able to convert into or replace each other. The activation of the MPs marker PDGF-α (platelet-derived growth factor receptor α), triggers several cellular responses, including cell migration, proliferation and differentiation. The CT-derived PDGF signaling was indeed identified in blastema during axolotl limb regeneration [52]. As for mammals, the relevance of Pdgfra^+^ MPs was also determined in the tissue restoration processes [53,54]. Most recent studies performed by Storer et al. [50] exposed that blastema is mainly formed by Pdgfra^+^ cells and that their progeny locate throughout the regenerating bone and dermis (Figure 4). Ultimately, ablation of the Pdgfra^+^ cells in the mouse model compromised the regeneration of the digit.

Several hypotheses were formed in order to explain the following regenerative mechanisms and determine the possible sources of the blastema cells. It was previously believed that these highly proliferative cells are all lineage-restricted, thus, are only the progenitors of their own tissue of origin [55,56]. Many challenged this concept, including Carr et al. [54], who studied nerve-derived mesenchymal cells in the blastema and identified their descendants in the dermis and bone of the regenerated digit. Other theories assumed the recruitment of the circulating precursors, which would transdifferentiate within the damaged tissue [52,57,58] and/or the involvement of the residual mature cells, which thenceforth dedifferentiated in the blastema [25,59,60,61,62,63]. The accumulation of such pluripotent class cells would further respond to different signaling pathways—other than those used in the development of digits and limbs. Indeed, it is currently under deliberation whether the blastema’s local environment affects the recruited cells’ phenotypes and, therefore, their contribution to the regenerating tissues. The single-cell RNA sequencing techniques (scRNA-seq), utilized by Storer et al. [50], provided new insights into the behavior, plasticity and heterogeneity of the CT progenitor cells within the blastema. The Pdgfra^+^ MPs were observed to acquire a unique plastic transcriptome that progresses and adapts over time as the blastema matures (Figure 4). The authors observed that the bone-lining Dmp1^+^ cells, in response to the digit tip injury, were capable of switching their mesenchymal lineage fate by depositing to both the bone and the dermis. Concurrently, the phenotypes of the dermal fibroblasts that were transplanted into the regenerating digits were considerably affected by the blastema environment and began to express the blastema-state genes. Ultimately, the injected fibroblasts contributed to the renewal of the bone. The same transplantation, performed in the proximally amputated, therefore non-regenerative digits, had no similar effect on the phenotype of the introduced cells, confirming the role of blastema in altering the inherent cells’ transcriptional state [50]. Cell population in the early blastema (11–17 days post-amputation) of the regenerating digit tip was recently analyzed via scRNA-seq by Johnson et al. [64], who identified the very same broad cell types in the intact unamputated digit tip. The most abundant and heterogeneous population within the blastema were fibroblasts, which were also observed to be enriched in several regeneration-specific markers, including a novel *Mest* gene [64]. The *Mest* gene expression was previously identified in mesenchymal tissues of the developing embryo [65].

In the early healing stages, the bone-forming regions are located alongside the periosteal surface of the remaining stump and are rich in actively proliferating cells that express markers for osteoblast commitment: Runx2, Sp7 and Osterix (OSX) [15,25,39,40]. The essential role of the periosteum in a successful digit tip regeneration was recognized by Dawson et al. [40], who showed that the tissue’s mechanical removal leads to a substantial reduction of the regenerated bone’s length and volume. Prior to the WE formation, the area connected with the periosteum forms a circumferential ring around the degrading bone. Only after the region merges with the endosteal/marrow compartment, the blastema proper is formed. At this point, the bone re-differentiation occurs, which is perceived as an ossification occurring distally to the stump [40]. Cells that participate in the process organize themselves through the extracellular matrix production, rich in Collagen III, which is subsequently degraded and replaced as the newly regenerated structures develop [39,66]. The process of ECM degradation and remodeling is promoted by matrix metalloproteinases (MMPs), activated by mechanical and chemical stimuli, triggered by the injury [67]. Initially, the restored bone, with an excessive volume at the dorsal-ventral axis, is rather trabecular in its nature, but ultimately it regains the general anatomy, length and tapered morphology of the native distal phalanx [42,44].

The healing process of the proximally amputated digit commences similarly to the regenerative response—with the inflammation and partial bone histolysis. However, the bone degradation is not as consistent and depends upon amputation level. Subsequently, cells that form the WE migrate across the top of the stump bone, separating the fibrin clot and completing the re-epithelialization within 13 days post-amputation [43]. At this point, fibroblastic cells, localized at the bone’s distal end, cover the stump with collagen fibrils and the cartilaginous callus forms annularly along the periosteal surface. Simultaneously, a dense collagen scar is formed in front of the bone stump. The callus’ ossification ultimately leaves the bone wider but shorter in length compared to the original amputation level. The Pdgfra^+^ MPs previously recognized to play a crucial role in blastema formation were also found in the non-regenerative environment and were observed to partially express blastema-associated genes [50]. However, these cells only contributed to the renewal of the bone stump. These results suggest that the inherent Pdgfra^+^ cells prime to partake in the regeneration but lack the decisive cues and signaling pathways that would reveal their plasticity and broad lineage potential.

### 5.1. BMP Signaling Play Crucial Role in Digit Regeneration

Based on several loss-of-function studies, it was determined that endogenous regeneration of the mammalian digit tips fails in the absence of several signaling pathways, including BMP, Wnt and growth factors secreted by the Schwann-lineage cells [15,21,50,53,54,68,69]. The crucial mediators of the BMP signaling, the Msx1-expressing cells, were previously identified within the nail bed structure [69]. Elimination of this tissue caused by the proximal amputation would explain why the site becomes regeneration incompetent. Although Msx1^+^ cells do not build the blastema, they are present in the distal clot, suggesting their role in blastema formation and maintenance [25]. It was additionally suggested that rather than the direct participation of Msx1^+^ descendants in the digit regeneration, the cells’ signaling properties are crucial for the process to succeed. In their studies, Han et al. [69] showed that digits of Msx1-deprived mice had downregulated expression of BMP4 and, therefore, displayed regeneration defects. This phenotype was reversed after administration of the exogenous BMP4. The activation of the BMP signaling pathway was also observed to be downregulated after inhibition of the transforming growth factor-beta (TGF-*β*). The TGF-*β* is required to initiate and regulate the regenerative response by modulating WE formation and blastema cell migration and proliferation [70,71,72]. Bone morphogenic proteins are recognized as potent proliferative molecules required for digit tip regeneration. Studies performed by Lee et al. [45] unveiled the BMP signaling role in the expression of the SDF-1α by the endothelial cells in the blastema. Moreover, it was shown that the proximal and thus regeneration-incompetent amputations of the digit can be induced by downstream mechanisms of the BMP signaling to undergo a segment-specific regenerative response [68,73,74,75,76]. The transition of a non-regenerative injury into the regeneration-competent wound was firstly demonstrated by Yu et al. [68], who treated the proximal amputation site with a single growth factor BMP7. Indeed, the treatment administered immediately after the amputation, performed on the neonatal mice, stimulated the ectopic bone formation [74,75]. However, the BMP7-induced response was distinct from the endogenous regeneration, as it involved a redevelopment of the wound and reactivation of differentiation programs, which then restored the amputated structure. In the following study, Yu et al. [76] unveiled that the BMP2 treatment can also induce the regeneration process of the proximal amputation, executed at the level of the second phalange. Analysis at the molecular level showed that the injection of BMP2-soaked beads into the proximally amputated digit resulted in a locally enhanced population of the SDF-1α and CXCR4 positive cells in the mesenchyme region between the bone stump and the bead [45]. The peak in responsiveness to the BMP2 treatment was correlated with the wound closure timing, determining a temporally specific regeneration window for activating the regenerating response through a BMP signaling pathway [49]. The most recent study published by Yu et al. demonstrated that the additional treatment with the BMP9 also stimulated the joint structure’s regeneration. Cells that responded to the treatment and contributed to the joint restoration would otherwise undergo a fibrotic differentiation in the non-treated, regeneration-incompetent injury [73].

### 5.2. WNT Signaling Consolidate Nail Mini Organ and Digit Regeneration

Mouse digit tip, amputated at its distal region, resumes its original structure within 5 months following the injury. However, in the absence of β-catenin, the Wnt signaling mediator, the digit fails to regenerate as expected, including a complete inhibition of the bone restoration [15,21,38]. As it was determined by Takeo et al. [15], in the conditional knockout mice models *K14^CreER^β-catenin^fl/fl^*, the proliferation of the Runx2^+^ progenitors and Sp7^+^ osteoblasts were repressed, the fibroblast growth factor (Fgf) signaling was downregulated and the shortened nerves were not reaching the regenerating site. The nerves distortion was affected by lack of semaphorin 5a (Sema5a), an axon-guidance molecule that should typically be upregulated through the Wnt signaling around 3rd-week post-amputation. In the absence of nerves within the regenerating nail structure, the nerve-dependent Fgf2 expression, induced by the Wnt activation, is blocked (Figure 4). Subsequently, the Runx2^+^ progenitors and Sp7^+^ osteoblasts proliferation is inhibited, and so the production of Bmp4 [15]. Similar suppression of blastema development was observed in control mice that were subjected to the surgical removal of the nerves [77,78]. The unnerved blastema is deprived of nerve-associated *Sox2*-positive dedifferentiated Schwann cell precursors (SCPs) and their secretions: platelet-derived growth factor AA (PDGF-AA) and Oncostatin M (OSM) (Figure 4) [53]. The absence of SCPs manifestly hinders the proliferation of blastema mesenchymal cells and bone regeneration [53,77]. Moreover, loss of nerves, which should provide a migration route for the mesenchymal precursors, prevents their deposition in the regeneration of mesenchymal tissues, bone and dermis [54].

While the distal amputation ensures digit regeneration, the extended injury results in a complete regeneration failure. The proximal amputation removes the visible nail plate, over 50% of the distal phalanx, the periosteum of the tip and the nail matrix’s distal region. Although a share of NSCs located in the proximal matrix is still at the site, the distal Wntless-expressing zone, required for the initiation of the Wnt signaling, is completely removed (Figure 4). Loss of both the Wnt-secreting NSCs and the Wnt-responsive mesenchymal periosteal cells ultimately blocks the terminal phalanx’s distal regrowth and the nail bed extension, thus leaving the amputation site regeneration-incompetent [38]. Interestingly, the stabilization of β-catenin in the K14+ epithelium and the nail matrix, after a completed re-epithelialization of the proximal wound in the *K14^CreER^β-catenin^fl/ex3^* transgenic mouse, contributed to enhanced expression of TCF1 in the area of regenerating distal matrix and the formation of nerves throughout the blastema. The innervation induced the expression of the Fgf2 and increased the proliferation of inherent Runx2^+^ mesenchymal cells, completing the digit bone regeneration (Figure 4). Alike stabilization of the β-catenin, performed in the amputation proximal to the nail matrix region, resulted in a failed organ regeneration. Lack of TCF1 expression indicated that the skin epidermis and the NSCs of the nail matrix respond differently to the presence of the β-catenin. As expected, the Runx2^+^ and Sp7^+^ cells identified in the mesenchyme of the proximal wound did not show proliferative activity [38].

The Wnt signaling was determined to strongly correlate with the expression domain Lgr6 in the nail matrix and the presence of β-catenin was identified in the nucleus of the Lgr-expressing NSCs. Lehoczky and Tabin [21] determined that the Lgr6^+^ cells accumulate in the wound blastema (in contrast to Lgr5^+^ cells), the regenerating bone and eccrine sweat glands. The animal model deprived of the Lgr6 receptors (*Lgr6^−/−^*) develops morphologically normal digits and nails, however, the regeneration of amputated tips was unsuccessful in over 12% of performed procedures. The structures that failed to regenerate properly were mainly characterized by disorganized and thick nail epithelium, invaded by dermal cells, which suggests the role of the Lgr6 receptor in the epithelial organization. Furthermore, the regenerated digit bones of the *Lgr6^−/−^* mice were significantly smaller than WT controls, supporting the authors’ theory that the descendants of the Lgr6-expressing cells could be involved in the formation of the ossification center at the distal part of the stump bone [21].

## 6. Concluding Remarks

The amputation of mice digits at the distal level, which ensures consequent complete regeneration, eliminates a quarter of the tip but preserves the nail epithelium, periosteum and the bone marrow [37,38,40]. The intact distal nail matrix, expressing Wntless, is required to consolidate the digit regeneration process by conducting the Wnt signaling in osteoblast and osteoclast precursors (Figure 4) [32]. Thus, it emphasizes the importance of nail distal matrix and nail stem cells in orchestrating the whole digit tip regeneration [15]. The regrowth of the severed mammalian tips is mainly controlled by the remaining tissue-specific reparation capacity, accompanied by the ossification processes, secured by the periosteum—a source of osteoprogenitor cells [40,49]. However, the inability to regenerate after suffering an extended injury was demonstrated to be overcome by a growth factor treatment, which promoted longitudinal bone regrowth, cavitation and chondrogenesis [68,73,74,75,76]. Nevertheless, the stimulation of such partial regenerative responses has not given rise to the wound epidermis formation nor the blastema, a structure that coordinates the cell recruitment and migration, differentiation, morphogenesis and pattern formation [15,38,40,41,42,45,46,48,53]. The profound comprehension of the blastema environmental signals that promote the acquisition of a mesenchymal blastema state could one day unveil the reasons for regeneration failures in mammalian injuries and provide insights into the therapeutic strategies of overturning these undesirable outcomes.

The first identified nail stem cell population of highly proliferative Ki67+ cells had been assigned to the nail matrix. Although these stem cells and their progenitors were described to contribute only to the nail plate formation, the nail stem cells also play a significant role in orchestrating digit tip regeneration by activating Wnt signaling in the distal nail matrix epithelium. The subsequent discovery of the slow-cycling NSCs, located in the basal layer of the NPF, unveiled a population of bifunctional cells, adaptive in response to wounding [20]. During homeostatic growth, these normally quiescent cells are accountable for the peri-nail region’s regenerative potential. However, upon injury, they hastily advance to deliver progeny to the nail matrix and the damaged nail structure. The plasticity of the NPFSCs and their contribution to different ectodermal organs show their high regenerative potential, which could be employed in the field of regenerative medicine. The procedure of engrafting a nail strip, which contained slow-cycling NPFSCs, has already shown promising results in restoring the nail units of immunocompromised murine donors [20]. In the future, we could envision the development of accurate techniques, which would allow for safe harvest and culture of human stem cells from both nail proximal fold and nail matrix. That would pave the way for the new treatments for skin, nail and digit defects, including digit amputation injuries. Ultimately, it would be the essential step towards entire limb restoration in humans.

The restoration of the full process of digit regeneration is a very challenging problem in medicine today. However, in the light of recent discoveries, it is emerging as a very promising research area. Future studies will require further understanding of complex interactions between the nail epithelium, including both populations of nail-organ stem cells, the periosteum, bone marrow, innervation and vasculature, resulting in the proper blastema formation.

## Figures and Tables

**Figure 1 ijms-22-02864-f001:**
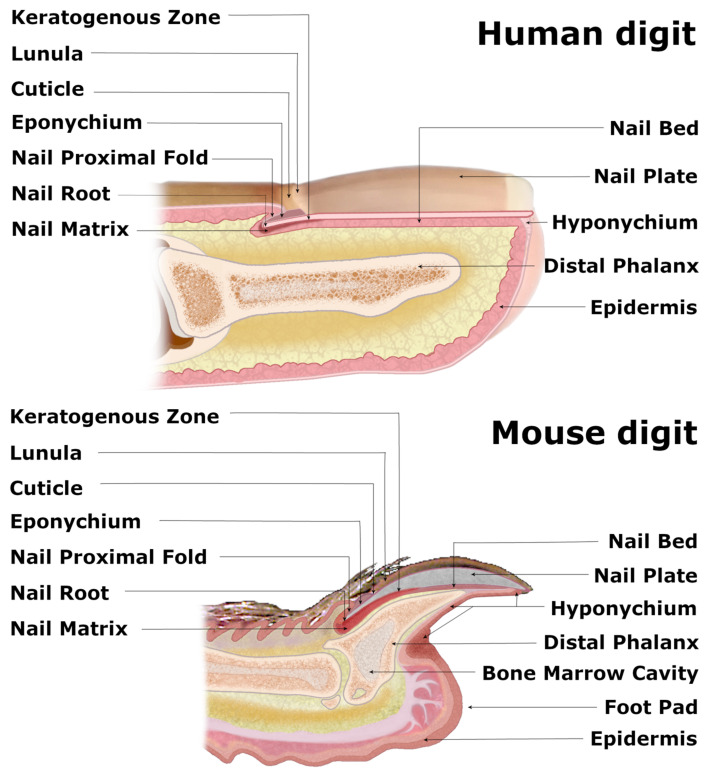
Comparative anatomy of human and mouse nail mini-organs. The visibly distinguishable nail organs of human and mouse digits share major characteristics across their inner structures. In both species, the wedge-shaped nail proximal fold (NPF), which smoothly transitions into the nail matrix (perceived as whitish crescent-shaped lunula), encircles the nail root of the nail plate (NP). Both flat human NP and claw-shaped murine NP lie atop of the keratogenous zone (KZ) and the nail bed (NB) and are protected with eponychium at the nail root and the hyponychium underneath the distal end of the NP. The triangular-shaped bone of the murine digit and splayed terminal phalanx of the human appendage enclose the bone marrow cavity.

**Figure 2 ijms-22-02864-f002:**
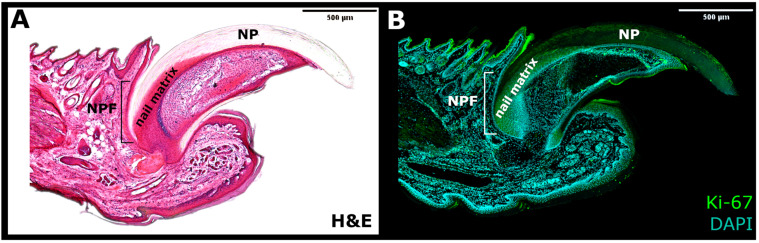
The side section of the nail mini-organ. (**A**) Haematoxylin and eosin (H&E) staining. (**B**) Immunofluorescence for Ki-67 marker of proliferation (Green) and DAPI (Blue). NPF—nail proximal fold; NP—nail plate.

**Figure 3 ijms-22-02864-f003:**
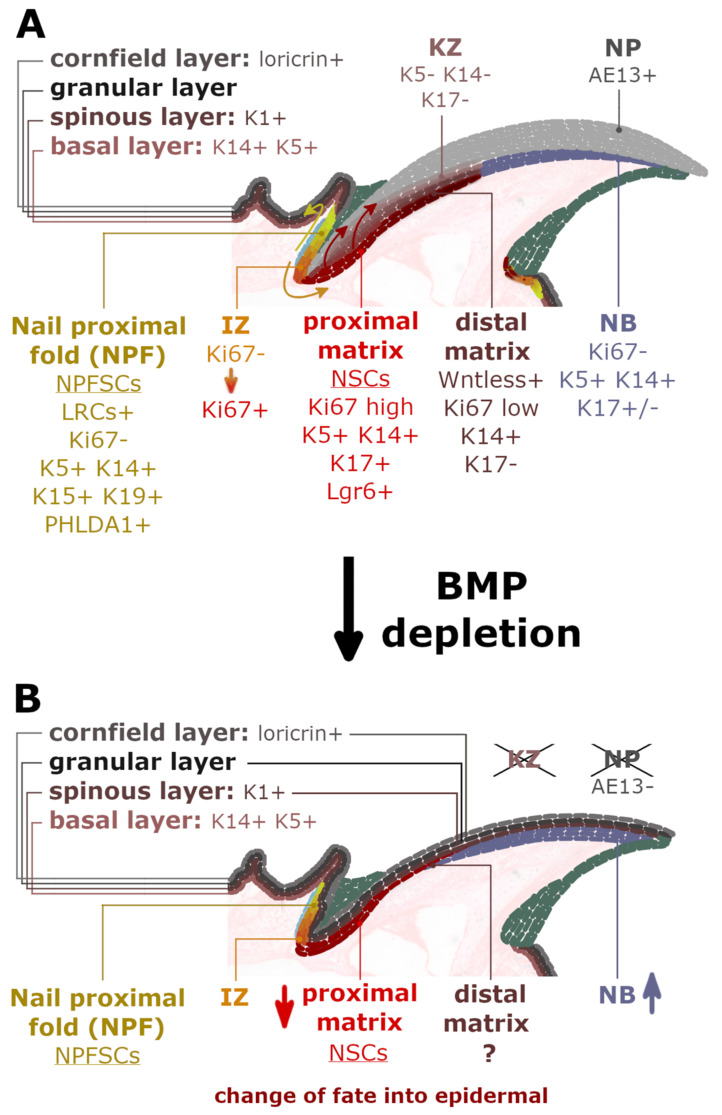
Nail organ markers and the requirement of the bone morphogenetic protein (BMP) signaling in proper nail development and differentiation. (**A**) Multi-layered skin epidermis expresses loricrin and soft keratins K1, K5, K14. The NPF region, rich in soft keratins K5, K14, K15 and K19, comprises slow-cycling LRC^+^ nail proximal fold stem cells (NPFSCs), proximally surrounded by the Lgr5 positive cells (in light blue). During normal homeostasis, the NPFSCs contribute long term to peri-nail epidermis (yellow arrow) but can also deliver the progeny to the nail matrix and differentiate into AE13-positive nail plate (NP) (orange and red arrows). The following intermediate zone (IZ) contains an intermediate gradient between slow- (Ki67-) to fast-cycling (Ki67+) cells, which reinforce the Lgr6-positive proximal matrix. Atop lies the keratogenous zone (KZ), devoid of the soft keratins. The distal matrix is an area of Wntless expression and thus the Wnt signaling pathway activation. This region’s activity was not yet determined in a mouse model deprived of the BMP signaling pathway. (**B**) The nail structures of the Bmpr1a-deficient mice manifested with an absence of the KZ and the hard NP structure, observed as a loss of the AE13 marker. Instead, the nail mini-organ was covered with a granular layer of extended skin epidermis, which expressed epidermal markers: loricrin and K1. Moreover, a significant reduction in matrix volume was observed and the hyperplasia of the NB.

**Figure 4 ijms-22-02864-f004:**
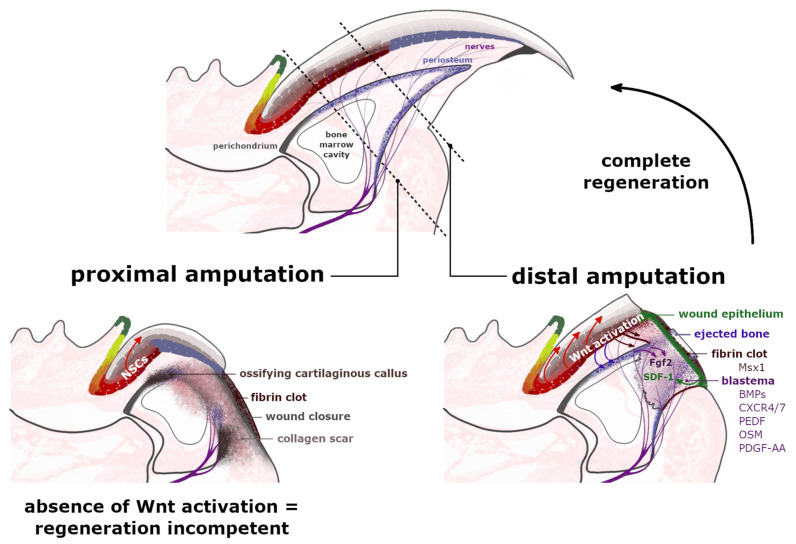
Regeneration of the amputated digit tips. Proximal amputation eliminates over 50% of the distal phalanx, removing the distal region of matrix epithelium, the periosteum, share of the bone marrow and the footpad while leaving the nails stem cells (NSCs) of the proximal matrix intact. Following the fibrin clot formation and the wound closure at the amputation level, a cartilaginous callus ossifies annularly along the bone surface, broadening the shortened stump along its dorsal-ventral axis. The formation of a dense collagen scar leaves the injured site regenerative-incompetent. The distal amputation, which eliminates up to 23% of the distal phalanx length, initiates the histolysis of the following 30% of the remaining stump. Between the ejected bone, protruded in a fibrin clot and an exposed bone marrow cavity, a transient wound epidermis (WE) forms, which through secretion of SDF-1 (green arrow) acts as a signaling center for CXCR4/7^+^ and Pedf^+^ cells. The WE outlines the blastema—an accumulation of highly proliferative cells, which express the bone morphogenetic proteins (BMPs) and antiangiogenic factor PEDF and ultimately contribute to the regenerating structures. The Msx1^+^ cells, present in the fibrin clot, do not build the blastema. The expression of Wntless protein by a distal nail matrix activates the Wnt signaling pathway, which promotes blastema innervation (burgundy arrows), which in turn provides the Fgf2 expression (purple arrows). Additionally, injured peripheral nerves provide a migration route for the Schwann cell precursors that secrete additional growth factors, including Oncostatin M (OSM) and platelet-derived growth factor AA (PDGF-AA). The presence of growth factors stimulates blastema cells to proliferate and differentiate. Furthermore, the active Wnt signaling interacts with periosteum (blue arrows) containing Wnt-responsive mesenchymal cells, which are crucial for the distal appositional bone growth and nail bed extension.

## Data Availability

Not applicable.

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
