# Peer review of "The Potential of Nail Mini-Organ Stem Cells in Skin, Nail and Digit Tips Regeneration"

_ijms, 2021, doi:10.3390/ijms22062864_

Round 1

Reviewer 1 Report

Dear authors, 

the review on The Potential of Nail Mini-Organ Stem Cells in Skin, Nail and Digit Tips Regeneration was nice to read. Although I am only in a distantly related field (biomaterials from the materials/chemistry point of view) I was able to learn from the review.  

Whether or not the review matches recent advancements in the field, I like to leave this to reviewers who are more familiar with the field. 

From my point of view only a few minor things should be addressed before considering publications. 

In continental Europe our languages allow very long sentences. Even some  beauty of poetry originates in the intricateness of sentences. Personally, it took me years to understand that English language mediates complexity as well as elegance on a different level. Namely, through thoughtful arrangement of short sentences. Nuances are often created by choices of word combinations rather than sentence structure.  

I like to share one advise that has served me very well. 

In case your writing has a sentence with more than 20 words, make two sentences out of it.  Sentences with 40+ words should definitely be restructured. 

The figures are very nice, but a bit busy. Do consider to increase intrinsic symmetry of the figures by aligning font size and text position to be more uniform. Alternatively, consider numbering within the image and listing of description on the right. 

Readers not fully embedded within the field, may appreciate a short significance statement at the end of each paragraph. Like: In summary, recent advancement abc indicate that xyz. 

Overall a very nice work that must have taken considerable effort to write. 

Author Response

   We sincerely thank all reviewers for the very helpful comments and suggestions. We are pleased to see very positive and constructive feedback from all reviewers suggesting only minor changes in our manuscript. We addressed all the questions providing detailed responses to all reviewers’ comments in file “Revisions”(in an attachment). 

In general, regarding comments from all reviewers, we improved the clarity by restructuring too long sentences. We thoroughly checked the whole text for English accuracy/spelling (all changes are marked in red in a revised version of the re-submitted manuscript). Regarding the figures, we modified almost all of them (Fig.1, Fig.3, and Fig.4), trying to increase intrinsic symmetry, aligning font size and position. We hope that figures are more uniform and less busy now. We also added short significance statements at the end of each section.

Regarding additional comments from reviewers, in the “Concluding Remarks” section, we added a paragraph about how we could envision translating these findings in a potential clinical setting in the future.

We have now addressed their comments and incorporated their suggestions in our revised manuscript and figures. In the attached file entitled “Revisions”, we delineated our responses to the reviewers’ specific comments.

We hope that our modifications of the manuscript further improved the quality of our work.

Sincerely,

Krzysztof Kobielak

Reviewer 2 Report

I think that this review ist of interest and worth to be published; there is only one minor open or missing. What ist the impact of differences between mouse and human or how can we translate these findings in a potential clinical seeting? This should be "discussed" in a more profound way (final remarks part) --> What is missing and what are the next steps!

Author Response

(The authors gave the same response as above.)

Reviewer 3 Report

1. good review from basic and clinical view points
2. Should the authors provide more information about application of two different nail mini-organ stem cells or interaction of nail and skin stem cells?

Author Response

(The authors gave the same response as above.)
